# Discovery of PANoptosis-related signatures correlates with immune cell infiltration in psoriasis

Li Wu[1,2‡], Xin-long Jiao[3,4‡], Ming Jing[5‡], Sheng-xiao Zhang[6], Yang Wang[7,8], Chen-long Li[1], Gao-xiang Shi[1,9], Zhuo-yang Li[7], Ge-liang Liu[7], Kai Yan[7,8,10], Li-xuan Yan[7,11], Qi Wang[1,7,8]*, Pei-feng He[8]*, Qi Yu[7,8]*

1 Department of Anesthesiology, Shanxi Provincial People's Hospital, Taiyuan, China, 2 School of Basic Medical Sciences, Shanxi Medical University, Taiyuan, China, 3 Academy of Medical Sciences, Shanxi Medical University, Taiyuan, China, 4 Department of Social Medicine, School of Public Health, Shanxi Medical University, Taiyuan, China, 5 Jinan Dermatosis Prevention and Control Hospital, Jinan, China, 6 Department of Rheumatology and Immunology, The Second Hospital of Shanxi Medical University, Taiyuan, China, 7 School of Management, Shanxi Medical University, Taiyuan, China, 8 Shanxi Key Laboratory of Big Data for Clinical Decision Research, Shanxi Medical University, Taiyuan, China, 9 Department of Anaesthesia, Shanxi Bethune Hospital, Shanxi Academy of Medical Sciences, Tongji Shanxi Hospital, Third Hospital of Shanxi Medical University, Taiyuan, China, 10 Department of Information Technology, Digital Health Guidance Center of Shanxi Province, Taiyuan, China, 11 Department of Anesthesiology, Second Hospital of Shanxi Medical University, Taiyuan, China

‡ LW, XJ and MJ are contributed equally to this work and share first author.
* yuqi@sxmu.edu.cn (QY); hepeifeng2006@126.com (PH); dr_wangqi_2020@163.com (QW)

**Data Availability Statement:** The datasets generated during and analyzed during the current study were available in the GEO database (https://

## Abstract

Psoriasis is an inflammatory skin disease that relapses frequently. Keratinocyte apoptosis dysregulation plays a crucial role in the pathological mechanisms of psoriasis. PANoptosis is a process with intermolecular interaction among pyroptosis, apoptosis, and necroptosis. The mechanism of PANoptosis in the occurrence and development of psoriasis is still unclear. Here we present a novel approach by identifying PANoptosis-related signatures (PANoptosis-sig) from skin tissue of psoriasis patients and healthy controls on transcriptional and protein levels. Five PANoptosis-sig (TYMP, S100A8, S100A9, NAMPT, LCN2) were identified. Enrichment analysis showed they were mainly enriched in response to leukocyte aggregation, leukocyte migration, chronic inflammatory response and IL−17 signaling pathway. Single cell transcriptome analysis showed TYMP and NAMPT were expressed in almost all cell populations, while LCN2, S100A8 and S100A9 were significantly highly expressed in keratinocyte. We then constructed predictive and diagnostic models with the PANoptosis-sig and evaluated their performance. Finally, unsupervised consensus clustering analysis was conducted to ascertain psoriasis molecular subtypes by the PANoptosis-sig. The psoriasis cohort was divided into two distinct subtypes. Immune landscape showed that the stromal score of cluster 1 was significantly higher than cluster 2, while the immune and estimate scores of cluster 2 were expressively higher than cluster 1. Cluster 1 exhibited high expression of Plasma cells, Tregs and Mast cells resting, while cluster 2 showed high expression of T cells, Macrophages M1, Dendritic cells activated, and Neutrophils in immune infiltration analysis. And cluster 2 was more sensitive to immune checkpoints. In conclusion, our findings revealed potential biomarkers and therapeutic targets for the

www.ncbi.nlm.nih.gov/geo), GeneCards database (https://www.genecards.org/), STRING database (https://string-db.org/).Proteome data of psoriasis was retrieved from the PRIDE database (https://www.ebi.ac.uk/pride). All data generated or analyzed during this study are included as supplementary information files.

**Funding:** This project was supported by grants from Science and Technology Innovation of Shanxi Province (No. 202304051001017), the National Social Science Fund of China (No. 21BTQ050), Shanxi Graduate Education Innovation Plan Project (No. 2023KY357) and Shanxi Basic Research Project (No. 202203021212361). There was no additional external funding received for this study.

**Competing interests:** The authors have declared that no competing interests exist.

prevention, diagnosis, and treatment of psoriasis, enhancing our understanding of the molecular mechanisms underlying PANoptosis.

## Introduction

Psoriasis is a prevalent chronic papulosquamous skin disorder, impacting a substantial population of over 600,000 adults and children globally [1]. The clinical manifestations of psoriasis are characterized by erythematous overlying silver-white scales and skin thickening [2]. The pathological mechanism of psoriasis is mainly characterized by abnormal differentiation and proliferation of keratinocytes, which is manifested by shortening of epidermal replacement time and cell cycle [3]. The dynamic balance between apoptosis and proliferation of keratinocytes in the skin maintains the homeostasis of the epidermis [4]. With the in-depth study of psoriasis, it has been found that there are different ways of death of keratinocytes in psoriasis skin lesions [5–7]. In addition to keratinocytes, locally infiltrated immune cells such as neutrophils also undergo cell death [8]. The expression of anti-apoptotic proteins such as *BCL-2* and *BCL-xL* in the lesions is up-regulated, suggesting that abnormal apoptosis of psoriatic keratinocytes may be involved in the occurrence and development of psoriasis [9].

PANoptosis, a distinctive innate immune inflammatory cell death pathway, is regulated by the PANoptosome complex in response to the perception of pathogens, pathogen-associated molecular patterns, damage-associated molecular patterns, or the production of downstream cytokines [10]. PANoptosome complexes function as molecular scaffolds for concurrently engaging multiple key molecules from the pyroptotic(NLRP3, ASC, Caspase-1), apoptotic (Caspase-8, Caspase-3, FADD), and necroptotic(RIPK1 and RIPK3) cell death pathways [11–15]. Genetic and biochemical evidence indicates that proteins within the PANoptosome can be broadly divided into three groups: (1) ZBP1 and NLRP3, which are likely PAMP and DAMP detectors, (2) ASC and FADD, serving as adaptors, and (3) CASP1, CASP8, RIPK1 and RIPK3, as catalytic effectors [16]. PANoptosis is a process with intermolecular interaction among pyroptosis, apoptosis, and necroptosis [17]. The discovery of PANoptosis provides a key to treating diseases, especially in the regulation of cell death. Research findings have demonstrated an association between the *IL-23/IL-17* axis and the programmed cell death signaling pathway in psoriatic lesions, which may contribute to extensive apoptosis and severe inflammatory responses [18, 19]. Extensive evidence has been provided to support the involvement of the *IL-17*-mediated *NLRP3* inflammasome complex in psoriatic inflammation [20, 21]. Additionally, the activation of the *NLRP3* inflammasome complex is closely linked to different forms of cell death, ultimately leading to heightened production of *IL-1β* and *IL-17* [22]. The heightened expression of necroptosis-associated molecules such as *RIPK1* and *RIPK3*, along with the activation of necroptosis facilitated by the *RIPK1/RIPK3* pathway, significantly contribute to the upregulation of *IL-17*, thereby inducing chronic inflammation in psoriasis [23, 24]. The correlation between psoriasis and various programmed cell death has been studied, however, the role of PANoptosis-a recently identified form of programmed cell death-in psoriasis remains underexplored.

In this study, we presented a novel approach by identifying PANoptosis-related signatures (PANoptosis-sig) from skin tissue of psoriasis patients and healthy controls at transcription and protein levels. Then used the PANoptosis-sig to classify distinct molecular subtypes of psoriasis, and the reliability of the typing was verified in the validation set. In addition, by analyzing the relationship between PANoptosis-sig and immune cell profiles, we provided new

insights into how PANoptosis may influence immune responses in psoriasis. Furthermore, we employed multiple machine learning algorithms to develop and validate diagnostic models based on PANoptosis-sig, marking a significant advancement in the precision of psoriasis diagnosis. This comprehensive analysis not only highlighted the relevance of PANoptosis-sig to various psoriasis subtypes, but also offered a fresh perspective on the pathogenic mechanisms underlying psoriasis. By elucidating the link between PANoptosis-sig and psoriasis aggressiveness, we aimed to contribute valuable insights that could improve therapeutic approaches and patient outcomes in psoriasis management.

## Methods

### Data collection and preprocessing

The transcriptome samples for the study were obtained from the Gene Expression Omnibus (GEO) database. Inclusion criteria encompassed a minimum sample size of 20, the simultaneous inclusion of both psoriasis patient and healthy control groups, and the tissue source of datasets derived from skin tissue. Five microarray datasets, three RNAseq datasets and one single-cell of 10× Genomics dataset were included for analysis (Table 1). Totally, 276 psoriasis lesion skin samples and 247 healthy control skin samples were obtained. The single-cell RNA (scRNA) dataset GSE151177 included 24, 354 cells, with the transcriptome profiles of 13 human psoriasis lesional skins selected for further analysis. GSE13355 was selected as the exclusive training set, the remaining datasets were employed as validation sets. We merged the datasets sequenced by microarray and RNAseq to remove batch effect respectively. Combat function in the R package "SVA" was used to correct the batch effects between the datasets after normalizing data using a log2 (TPM + 1) transformation [25]. The effectiveness of batch effect removal was evaluated using the Principal Component Analysis (PCA) method.

Proteome data of psoriasis was retrieved from the PRIDE database (http://www.ebi.ac.uk/pride). The inclusion criteria for the dataset were as follows: 1) including both psoriasis and normal samples. 2) derived from adult human skin tissue. Finally we found the dataset PXD021673 which contained samples from 5 patients with psoriasis and 5 healthy controls for further analysis. The quantification of label-free proteins was performed with MaxQuant software version 2.6.3.0 [26]. The human protein sequences from the Uniprot database (https://www.uniprot.org/) were used to search the peak lists. Both proteins and peptides with a minimum length of seven amino acids had a false discovery rate (FDR) of 0.01. The mass deviation of the precursor was allowed up to 20 ppm and that of the fragments was allowed up to 20 ppm during peptide identification.

**Table 1. The data series used in study.**

| Data Set | Platforms | Psoriasis/Responds | HC/No-Responds | Sample Type |
|----------|-----------|--------------------|----------------|-------------|
| GSE13355 | GPL570 | 58 | 64 | Skin-tissue |
| GSE14905 | GPL570 | 33 | 21 | Skin-tissue |
| GSE78097 | GPL570 | 27 | 6 | Skin-tissue |
| GSE109248 | GPL10558 | 17 | 14 | Skin-tissue |
| GSE53431 | GPL10558 | 12 | 12 | Skin-tissue |
| GSE54456 | GPL9052 | 90 | 81 | RNA-seq |
| GSE66511 | GPL16288 | 12 | 12 | RNA-seq |
| GSE121212 | GPL16791 | 27 | 37 | RNA-seq |
| GSE151177 | GPL18573 | 13 | 5 | Sc-RNA seq |
| PXD021673 | - | 5 | 5 | Proteomic data |

## Weighted gene co-expression network analysis (WGCNA)

The analysis workflow of WGCNA involved the utilization of the R package "WGCNA" to construct an scalefree network using the expression matrix. Subsequently, the Topological Overlap Matrix (TOM) was constructed to capture the potential regulatory network between genes. Hierarchical clustering was then performed on the TOM matrix, resulting in the generation of a clustering tree. Cutting the tree leads to the identification of distinct expression modules, with a focus on modules containing more than 50 genes for the purpose of identifying gene modules associated with psoriasis. Furthermore, the merging of similar modules facilitated the construction of an unsupervised co-expression network. Various branches of clustering analysis demonstrated separate gene modules, wherein genes within the same module exhibited a greater level of co-expression, while co-expression between genes in different modules was comparatively lower. Modules displaying higher correlation were suggestive of genes closely linked to the target disease [27].

## Identification of differentially expressed genes in psoriasis

Normalization and log2 conversion were performed to identify differentially expressed genes (DEGs) in the GSE13355 dataset. We employed the R package "limma", which applied a generalized linear model technique to detect DEGs [28]. DEGs selected based on the criteria of |log2 fold change| > 1.5 and *adj p*-value < 0.05. The "ggplot2" package was used for the generation of volcano plots for the DEGs [29].

## Identification of differentially expressed proteins in psoriasis

The downstream data of LC-MS were analyzed by perseus(version 2.1.1.0) [30]. The analysis excluded protein groups identified only by site, peptides identified in the reverse database, and those belonging to the common contaminants database. The missing values were imputed with an imputed width of 0.3 and a downshift of 1.8 in Student's t-test. With S0 set to 0, two sample tests were run. A minimum ratio count of 1 was used for label-free quantification. Proteins were considered differentially expressed based on the criteria of |log2 fold change| > 1 and *adj p*-value < 0.05.

## Identification of PANoptosis-related signatures in psoriasis

A total of 1325 genes (1313 apoptosis genes, 11 necrosis genes, and 31 pyroptosis genes) were obtained by filtering PANoptosis-related genes from the GeneCards database, using a relevance score > 3 (S1 Table). The intersection of genes among PANoptosis, DEGs and the most relevant module genes from WGCNA were designated as PANoptosis-related DEGs(PDEGs). The intersection of proteins between PANoptosis and differentially expressed proteins(DEPs) were designated as PANoptosis-related DEPs (PDEPs). The PANoptosis-related signatures (PANoptosis-sig) were defined as the intersection of PDEGs and PDEPs.

The friends analysis method was utilized to ascertain genes that exhibited a stronger correlation with others within the same pathway, thereby suggesting their potential significance in regulatory functions. The friends analysis approach examined the direct correlations among PDEGs, assigning a total score between 0 and 1, where a higher score indicated a stronger association with other genes. To investigate the interaction relationships among PDEGs, the STRING database (https://string-db.org/) [31] was employed. A confidence score exceeding 0.4 was established as the threshold for constructing the protein-protein interaction (PPI) network.

## Single-cell RNA sequencing data analysis

In order to investigate the relationship between PANoptosis-sig and immune cell, we analysed single-cell RNA sequencing of emigrating cells from human psoriasis skin. R package "Seurat" (4.3.0) was used to analyze and cluster downstream single-cell data [32]. Data quality control was performed separately for each sample before data integration. In order to eliminate partial cells and doublets, genes expressed in fewer than three cells, cells with more than 5,000 genes, and mitochondrial genes expressed in more than 25% by mitochondrial cells were filtered out. It was then necessary to create Seurat objects, scale data, and find variable 2,000 genes. Graph-based clustering and PCA were performed with "Seurat" package. We selected twenty principal components for uniform manifold approximation and projection. FindClusters function was used to cluster the cells. AverageExpression function was used to calculate average gene expression for psoriasis within clusters. We excluded clusters expressing two or more gene signatures of different immune cells from downstream single-cell analysis to avoid clusters with possible doublets.

## PANoptosis-sig based diagnostic model development and validation

A series of 14 commonly used machine learning algorithms including logistic regression(LR), linear discriminant analysis(LDA), quadratic discriminant analysis(QDA), k-nearest neighbor (KNN), decision tree(DT), random forest(RF), xgboost, lasso regression, ridge regression(RR), elastic net regression(ENR), stepwise logistic regression, support vector machine(SVM), grandient boosting machine(GBM) and naive bayesian algorithm were used to develop diagnostic prediction model for the systematic use of skin to tissues based on PANoptosis-sig. We used the combination methodology to construct the final classification model based on the training set and validation sets. The model performance was validated using comprehensive composition validation techniques. The predictive performance of the models were evaluated by calculating the area under the ROC curve(AUC) using the "pROC" package.

## Identification of psoriasis subtypes driven by PANoptosis-sig

In order to gain a deeper understanding of the heterogeneity of PANoptosis-related molecular subtypes in psoriasis, we employed consensus clustering using the "ConsensusClusterPlus" R package. The parameters were set as follows: maximum k = 6, number of repetitions = 1000, pitem = 0.8, pfeature = 1, clusteralg = hc, distance = pearson [33]. The optimal cluster assignment was determined by conducting 1000 iterations of the hc algorithm and utilizing the cumulative distribution function (CDF). PCA was utilized to visually represent the disparities among the subtypes in a scholarly manner.

## Functional enrichment analysis

In order to ascertain pathway enrichment among the DEGs of GSE13355, PDEGs, PANoptosis-sig and the DEGs between the molecular subtypes of psoriasis, Gene Ontology (GO) functional enrichment analysis and Kyoto Encyclopedia of Genes and Genomes (KEGG) pathway enrichment analysis were conducted. Significantly enriched functional pathways were determined based on adj $p$-value $< 0.05$ [34].

## Immune microenvironment analysis between different psoriasis subtypes

In this study, we compared immune scores between different psoriasis subtypes using "estimate" package, including StromalScore, ImmuneScore, and ESTIMATEScore. CIBERSORT was a computational tool that employed a support vector regression model trained on gene

expression data from purified immune cell populations to estimate the abundance of various immune cell types within a heterogeneous cell population [35]. CIBERSORT was utilized for immunocell analysis to deconvolute the composition of immune cells in complex tissue samples. The resulting immune infiltration data was visualized using the "ggpubr" package, with a significance threshold of $p < 0.05$. Additionally, immune checkpoints-immunoinhibitor, immunostimulator and major histocompatibility complex (MHC) analyses were employed to investigate the interaction between different specific subtypes. Pearson correlation analysis was utilized to investigate the association between PANoptosis-sig and immune cells.

### Statistical analysis

All statistical analyses were conducted using R software (version 4.1.3; https://www.r-project.org/). We analyzed the differences between two groups using t-test and wilcox test. All statistical tests were two-sided, and significance was determined as $p < 0.05$.

## Results

### Identifying differentially expressed genes (DEGs) in psoriasis

In GSE13355, a total of 256 DEGs were identified from psoriasis and healthy control samples, with 191 genes upregulated and 65 genes downregulated (Fig 1A, S2 Table). The heatmap depicted the top 30 genes that were upregulated and downregulated (Fig 1B). The analysis of GO functional enrichment demonstrated significant connections between these DEGs and various processes, including cytokine-mediated signaling pathways, skin development,

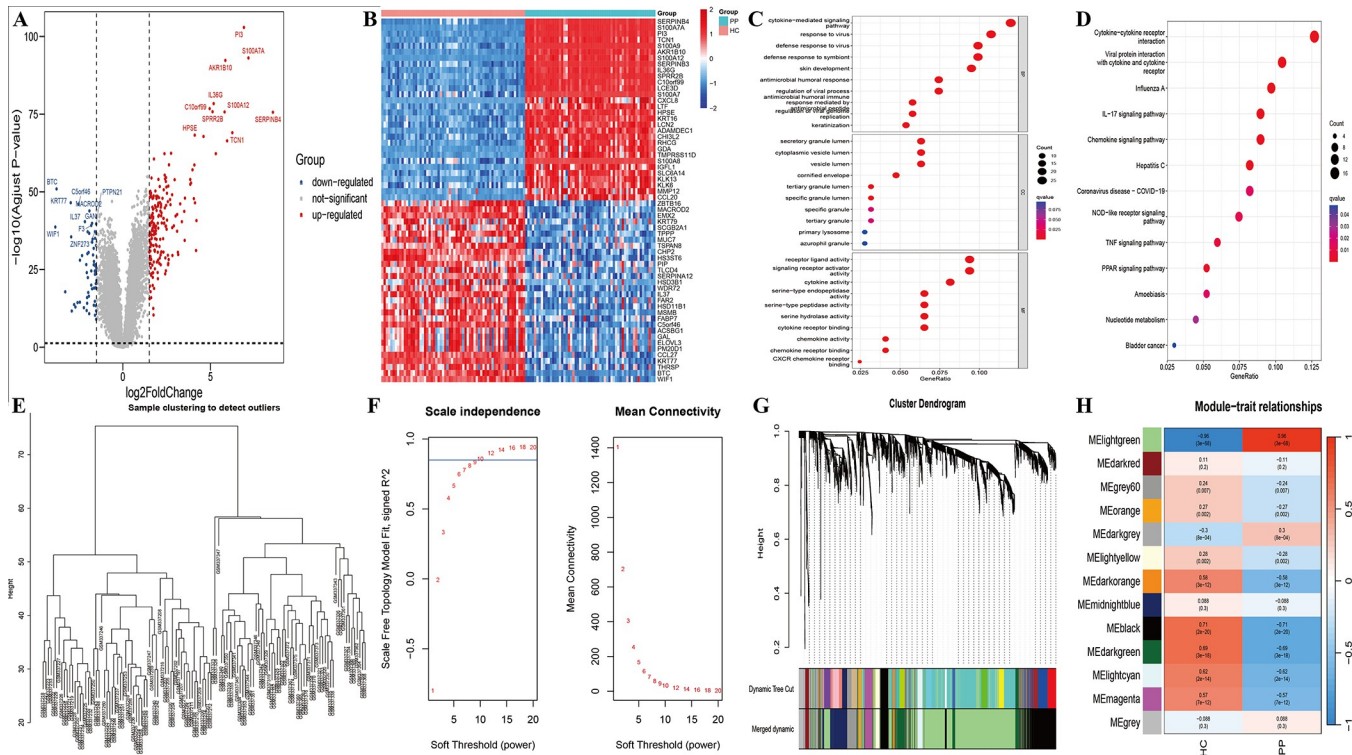

**Fig 1. Identification of DEGs and WGCNA analysis in GSE13355.** (A) Volcano plot depicting DEGs between psoriasis patients and healthy controls. (B) Heatmap of DEGs. (C) GO enrichment analysis of DEGs. (D) KEGG enrichment analysis of DEGs. (E) Sample clustering dendrogram. (F) Optimal soft threshold power. (G) Cluster dendrogram of merged similar modules. (H) Heatmap of module-trait correlations.

replication, and keratinization (Fig 1C). Additionally, the KEGG pathway enrichment analysis indicated notable enrichment in pathways associated with cytokine and cytokine receptor interaction, *IL-17* signaling, Nod-like receptor signaling, *TNF* signaling, and nucleotide metabolism (Fig 1D).

### Identification of core module genes in psoriasis through WGCNA

The WGCNA method was utilized to ascertain the most pertinent module genes between the psoriasis and normal groups. The dataset GSE13355 encompassed 58 psoriasis samples and 64 healthy control samples. The examination of the sample clustering tree demonstrated no notable aberrations (Fig 1E). To establish the soft threshold, a scale-free topology criterion was employed, resulting in the observation of a scale-free gene network at a soft threshold power (β) of 10 (SFT.R.sq = 0.858) (Fig 1F). Subsequently, the modules were merged for further investigation using the parameters minModuleSize = 25 and MEDissThres = 0.25 (Fig 1G). Ultimately, the WGCNA analysis successfully identified 13 modules. The lightgreen module demonstrated significant positive correlation with psoriasis (r = 0.96, p = 3e-68), whereas the black module displayed notable negative correlation (r = -0.71, p = 2e-20) (Fig 1H). These two modules were identified as the most pertinent to psoriasis.

### Identifying differentially expressed proteins (DEPs) in psoriasis

In PXD021673, a total of 366 DEPs were identified from psoriasis and healthy control samples, with 300 proteins upregulated and 66 proteins downregulated (Fig 2A, S3 Table). Fig 2B showed a heatmap of all differentially expressed proteins in each sample.

### Identifing of PANoptosis-sig in psoriasis

We conducted an analysis that involved the intersection of DEGs, PANoptosis and the two modules with the highest positive and negative correlations from the WGCNA. This analysis resulted in the identification of 36 PDEGs (Fig 2C, S4 Table). The relationships among PDEGs

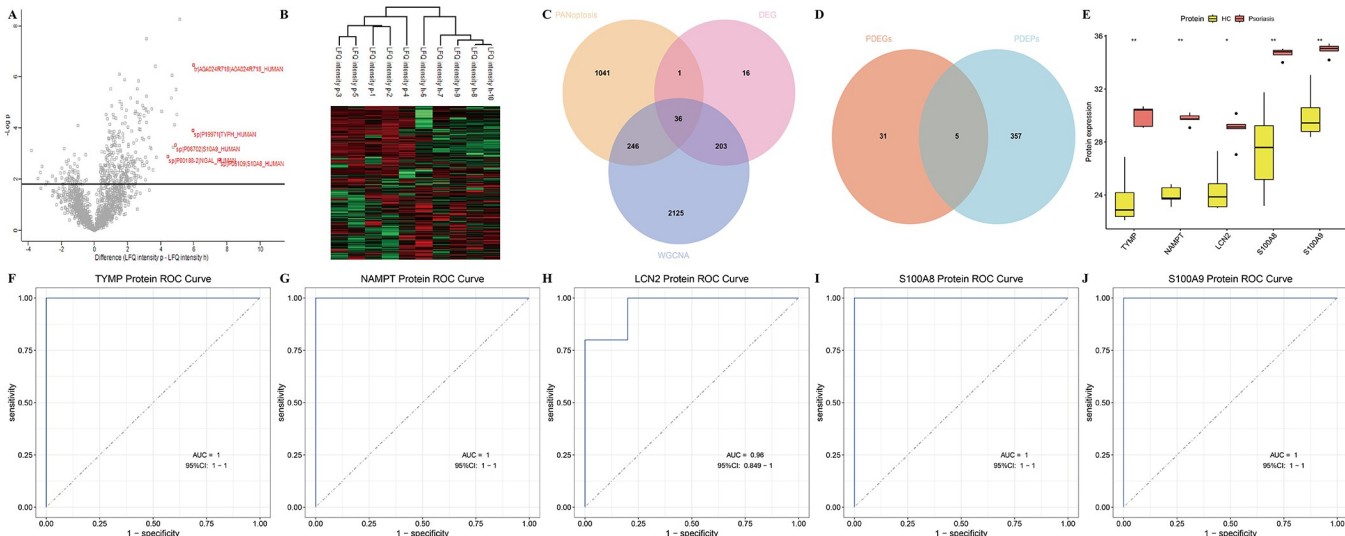

**Fig 2. Identification of PANoptosis-sig.** (A) Volcano plot depicting DEPs between psoriasis patients and healthy controls. (B) Heatmap of DEPs. (C) Venn diagram depicting the intersection of DEGs, PANoptosis-related genes, and crucial module genes from WGCNA. (D) Venn diagram depicting the intersection of DEGs and DEPs. (E) The expression of PANoptosis-sig between psoriasis patients and healthy controls in proteome. (F-J) ROC curves displaying the AUC values in proteome.

were investigated through PPI network and friends analysis. The PPI results and friends analysis demonstrated strong correlations between *S100A9*, *S100A8*, *NAMPT*, *TYMP* and others (S1A and S1B Fig), suggesting their potential involvement in the regulation of PANoptosis in psoriasis. Furthermore, GO functional enrichment analysis revealed significant enrichment in biological processes such as response to lipopolysaccharide, leukocyte migration, and chemotaxis (S1C Fig). The KEGG pathway enrichment analysis revealed significant enrichment in pathways associated with cell apoptosis, *IL-17* signaling, and chemokine signaling (S1D Fig). These findings provided evidence for the association between psoriasis and immune inflammation, suggesting the pivotal involvement of PANoptosis in the development of psoriatic inflammation.

Five PANoptosis-sig (TYMP, S100A8, S100A9, NAMPT, LCN2) were intersected by PDEGs and PDEPs (Fig 2D, S5 Table). Boxplots showed their significantly differential expression between psoriasis and healthy controls at the proteome level (Fig 2E). Area under the ROC curve (AUC) values of PANoptosis-sig curves were 100%, except LCN2(AUC = 0.96) (Fig 2F–2J). The results indicated that PANoptosis-sig had high potential for use as diagnostic biomarkers for psoriasis on the proteome level.

## Validating the expression of PANoptosis-sig in psoriasis

The expression of PANoptosis-sig were confirmed in both microarray and RNAseq datasets at the transcriptional level. In combined microarray datasets, box plots demonstrated significant upregulation in the psoriasis group (Fig 3A). ROC analysis indicated excellent diagnostic performancewith AUC values (TYMP = 0.99, NAMPT = 0.943, LCN2 = 0.987, S100A8 = 0.991, S100A9 = 0.999) (Fig 3C–3F). In RNAseq datasets, box plots revealed significant upregulation in the psoriasis group (Fig 3G). ROC analysis showed AUC values (TYMP = 0.996, NAMPT = 0.977, LCN2 = 0.987, S100A8 = 0.999, S100A9 = 0.999)97 (Fig 3H–3L). Collectively, these findings indicated that PANoptosis-sig had high predictive performance in the diagnosis of psoriasis on the transcriptome level.

## The correlation analysis between PANoptosis-sig and immune cells in psoriasis

The pearson correlation between PANoptosis-sig and immune cells was analyzed. It was noteworthy that PANoptosis-sig exhibited positive correlations with macrophage, DC, MSC, Th1 cells, Th2 cells, melanocyte, monocyte, neutrophil, basophil, sebocytes, epithelial and keratinocytes, negative correlations with HSC, adipocyte, endothelial, fibroblasts and mast cells (Fig 4).

## Expression distribution of PANoptosis-sig in single-cell RNA sequencing

Based on data reduction analysis of 15,497 single cells, clusters of NK cells, CD4 T cells, CD8 T cells, CD161 T cells, Treg cells, mature DCs, semimature DCs, macrophage, melanocyte, and keratinocyte (KCs) were identified at different layers of stratum corneum, stratum spinosum, and stratum basal (Fig 5A, S2 Fig). Subsequently, we identified the distribution of PANoptosis-sig in different cell populations (S3 Fig). The results further confirmed that *TYMP* and *NAMPT* were expressed in almost all cell populations(Fig 5B and 5C), while *LCN2*, *S100A8* and *S100A9* were significantly highly expressed in KCs (Fig 5D–5F).

## PANoptosis-sig based diagnostic model development and validation

To determine whether the PANoptosis-sig can build a cross-organizational, comprehensive, predictive model for psoriasis, we combined multiple machine learning methods. The AUCs

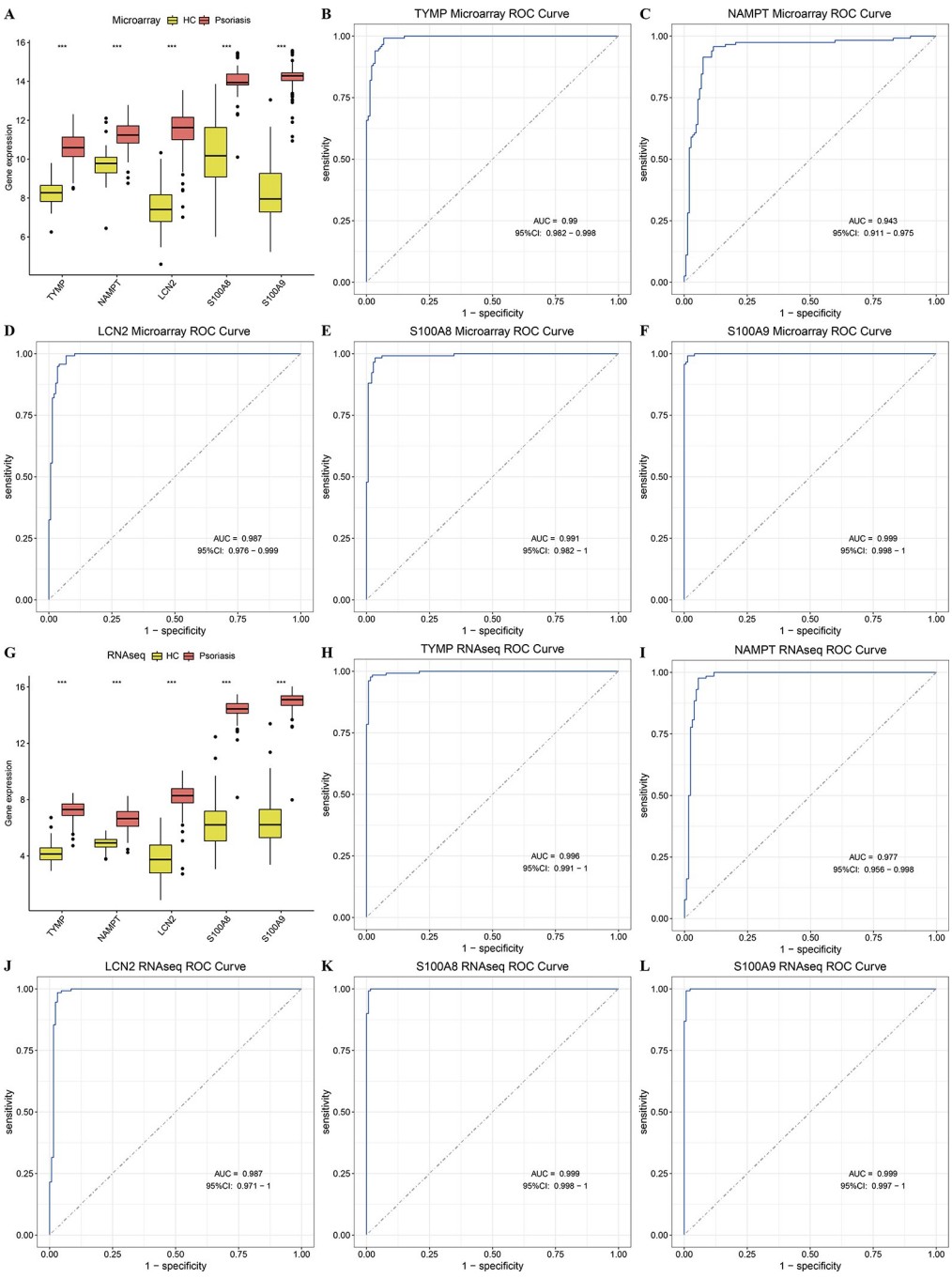

**Fig 3. Expression of PANoptosis-sig in microarray and RNAseq datasets.** (A) The expression of PANoptosis-sig in combined microarray datasets. (B-F) ROC curves displaying the AUC values in combined microarray datasets. (G) The expression of PANoptosis-sig in combined RNAseq datasets. (H-L) ROC curves displaying the AUC values in combined RNAseq datasets.

of the machine learning models were averaged across cohorts. Most predictive models in the training set showed outstanding diagnostic performance (AUCs = 1). We found that for RNA-seq data, the diagnostic models performance of XGBoost and GBM were superior, while the ENR model demonstrated better effectiveness for microarray data (Fig 6A). Based on the results, the PANoptosis-sig models could efficiently detect patients with or without psoriasis

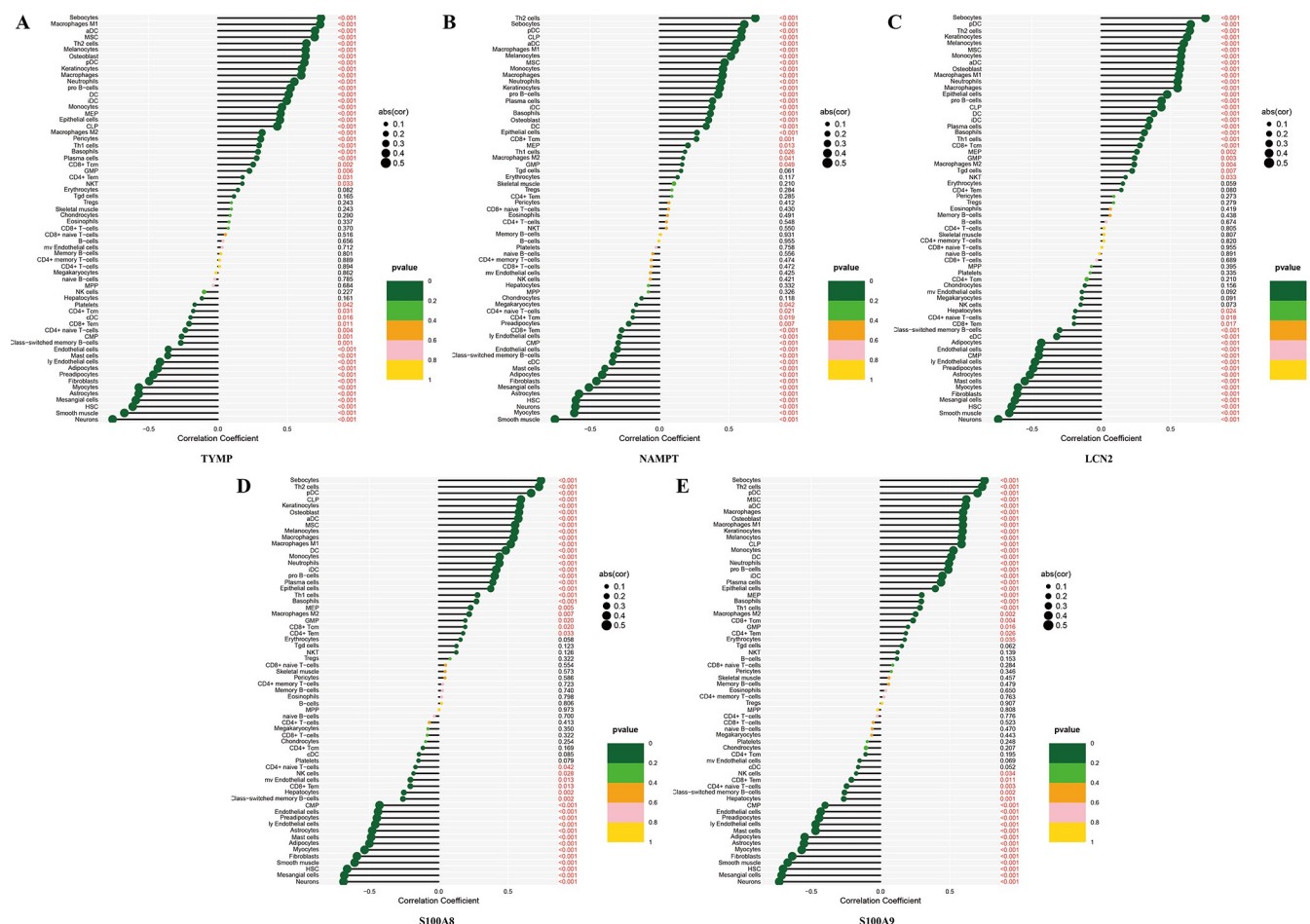

**Fig 4. Correlation analysis between PANoptosis-sig and immune cells.** (A-E) Correlation analysis between TYMP, NAMPT, LCN2, S100A8, S100A9 and immune cells.

and optimized clinical decision-making. In addition, we performed GO and KEGG enrichment analyses of PANoptosis-sig. The results showed PANoptosis-sig were mainly enriched in response to leukocyte aggregation, leukocyte migration, chronic inflammatory response and *IL−17* signaling pathway (Fig 6B and 6C). The results suggested the occurrence and progression of psoriasis were closely related to infection and inflammation.

## Clustering of different subtypes driven by PANoptosis-sig in psoriasis

This study employed the combat algorithm to mitigate batch effects in microarray data obtained from skin tissues of 147 psoriasis patients across five datasets (GSE13355, GSE14905, GSE78097, GSE109248 and GSE53431). The application of PCA visualization revealed noticeable alterations pre and post batch effect correction (Fig 7A and 7B). Furthermore, unsupervised consensus clustering analysis was conducted to ascertain psoriasis molecular subtypes. Upon setting k = 2, psoriasis was categorized into two distinct subtypes, with PCA analysis confirming the distinct separation of the subtypes (Fig 7C–7F). Heatmap displayed the expression of PANoptosis-sig in two subtypes as depicted in Fig 7G. The differential gene expression analysis between the two subtypes yielded 250 upregulated genes and 111 downregulated genes (S6 Table). Subsequently, GO and KEGG enrichment analyses were conducted on the

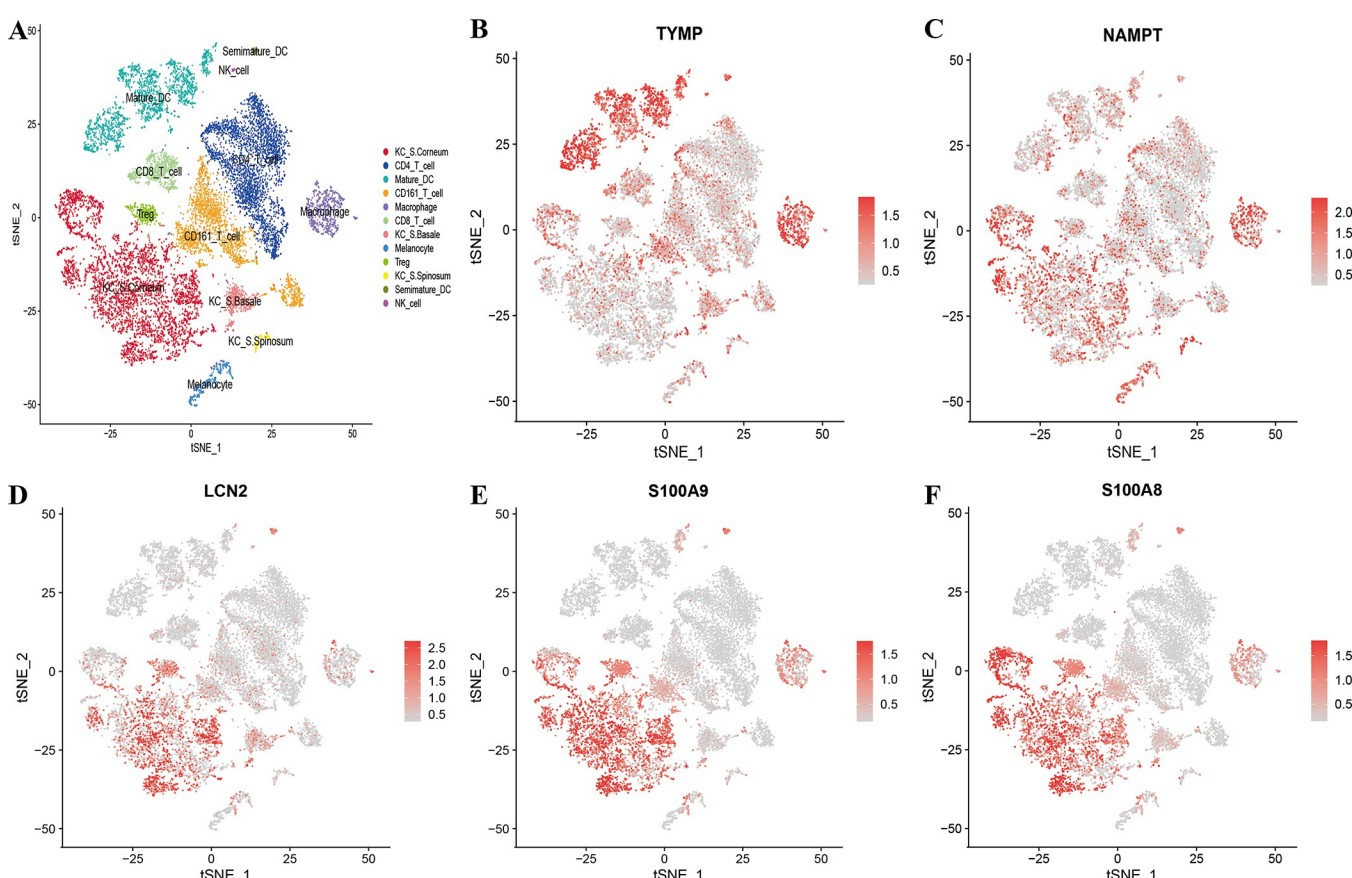

**Fig 5. Expression distribution of PANoptosis-sig in GSE151177.** (A) Cell classification of scRNA-seq data presented in the t-Distributed Stochastic Neighbor Embedding plot of psoriasis skin. (B-F) Expression distribution of TYMP, NAMPT, LCN2, S100A8, S100A9 in GSE151177.

361 DEGs. The results of the GO functional enrichment analysis demonstrated significant associations with various biological processes, including cytokine-mediated signaling pathways, epidermal development, skin development, leukocyte migration and keratinocyte differentiation (Fig 7H). Additionally, the KEGG pathway enrichment analysis revealed significant enrichment in pathways such as cytokine-cytokine receptor interaction, *IL-17* signaling pathway, and chemokine signaling pathway (Fig 7I).The outcomes of the enrichment analyses revealed significant enrichment of immune and inflammation-related functions once again. The reliability of the PANoptosis-sig-based molecular subtyping had been validated in RNA-seq datasets, demonstrating the stability of subtyping in psoriasis (S4 Fig).

## Immune landscape between psoriasis subtypes

In order to examine specific differences in the immune microenvironment between psoriasis subtypes, we compared stromal, immune and estimate scores. The results showed that the stromal score of cluster 1 was significantly higher than that of cluster 2, while the immune and estimate scores of cluster 2 were expressively higher than those of cluster 1 (Fig 8A–8C). The levels of immune cell infiltration were examined between two subtypes. Cluster 1 exhibited high expression of Plasma cells, Tregs and Mast cells resting, while cluster 2 showed high expression of T cells, Macrophages M1, Dendritic cells activated, and Neutrophils (Fig 8D). Then the expression of immune checkpoints were analyzed between the subtypes. Cluster 2 was characterized by high expression levels of most of the MHC (Fig 8E), immunoinhibitor

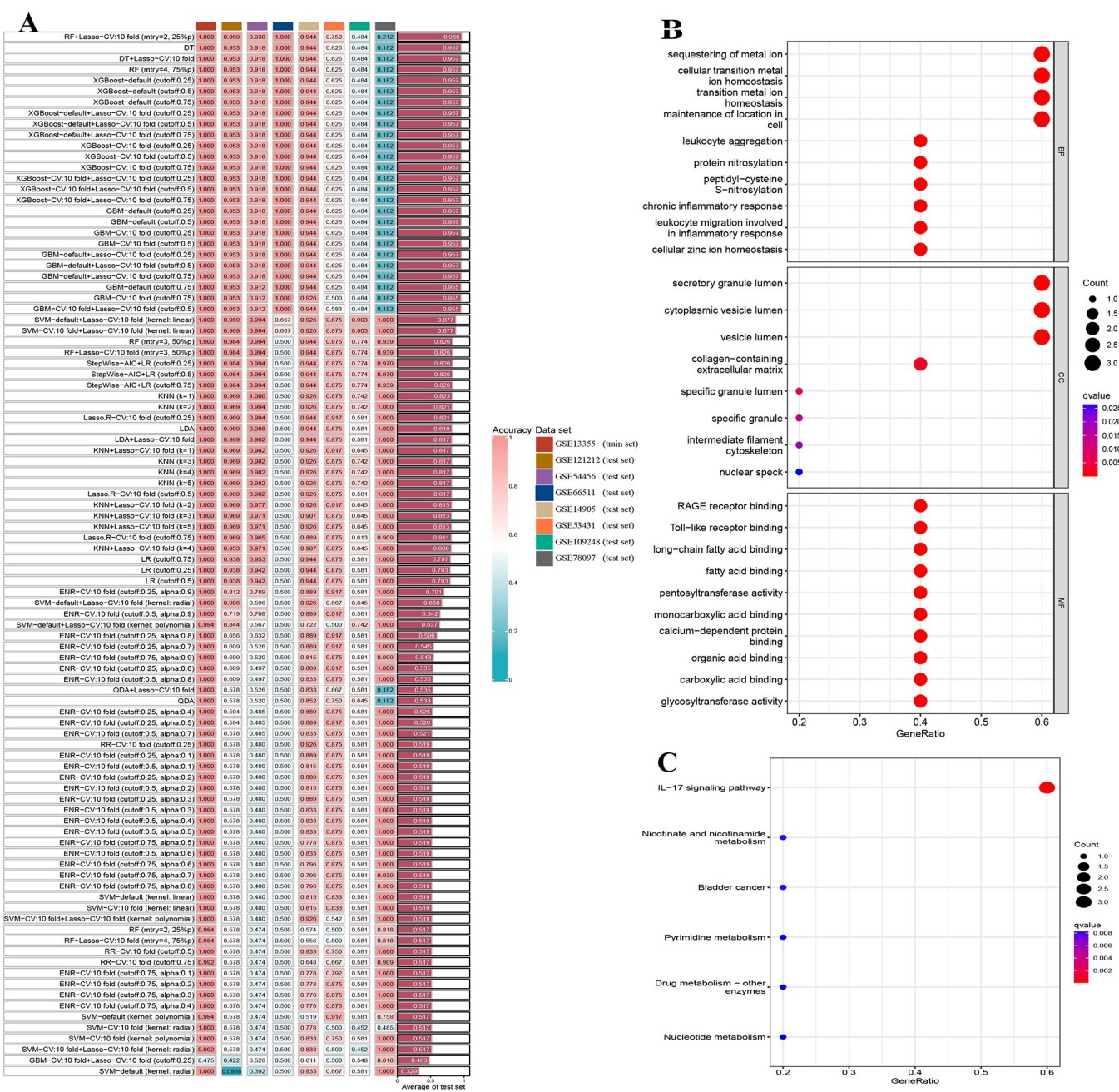

**Fig 6. Construction of the prediction model by PANoptosis-sig.** (A) The AUC value of multiple machine-learning algorithm combinations in eight cohorts. (B) GO and KEGG (C) enrichment analysis of the PANoptosis-sig.

(Fig 8F), immunostimulatory checkpoints (Fig 8G). There might be a relationship between PANoptosis-related subtypes and immunotherapy effectiveness, indicating that cluster 2 displayed heightened sensitivity to immunotherapy.

## Discussion

The normal homeostasis of psoriatic epidermal cells was mainly based on the normal mechanism of apoptosis [36]. Research had been conducted on how different treatment methods for

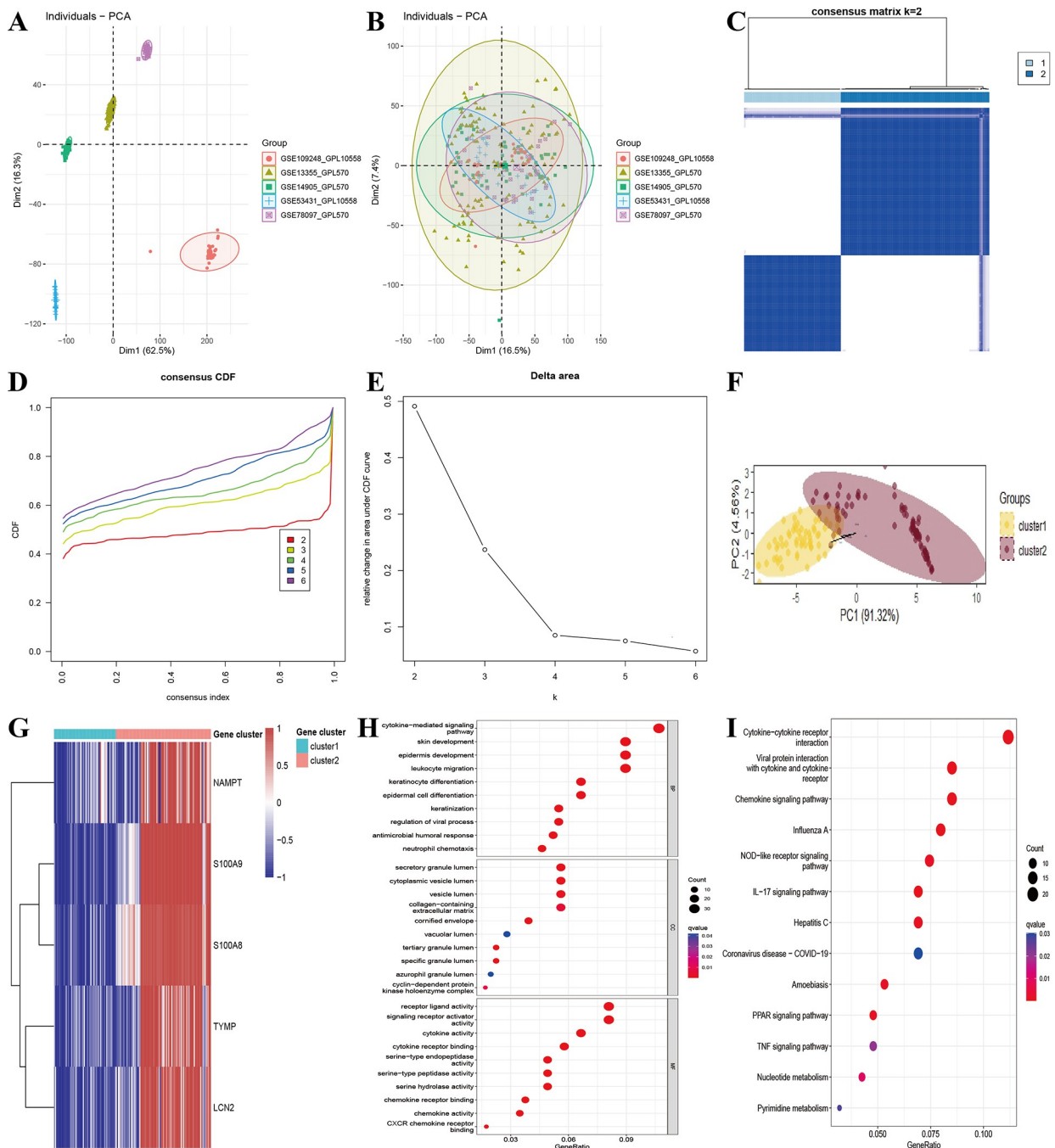

**Fig 7. Identification of psoriasis subtypes.** (A) Principal component analysis of expression matrices from 5 different datasets before batch correction. (B) Principal component analysis of expression matrices from 5 different datasets after batch correction. (C) Consensus matrix heatmap when k = 2. (D) Cumulative Distribution Function (CDF) of consensus clustering. (E) Relative changes in Delta Area under the CDF curve. (F) Principal Component Analysis (PCA). (G) The expression of PANoptosis-sig in two subtypes. (H-I) GO and KEGG enrichment analysis of DEGs between the two subtypes.

psoriasis affected keratinocyte homeostasis and cell death regulation for the past few years [5, 37]. PANoptosis, a form of cell death, was closely related to immune and inflammatory response and played a vital role in the pathogenesis of psoriasis [38]. A better understanding of PANoptosis and its underlying mechanisms in psoriasis was essential for improving

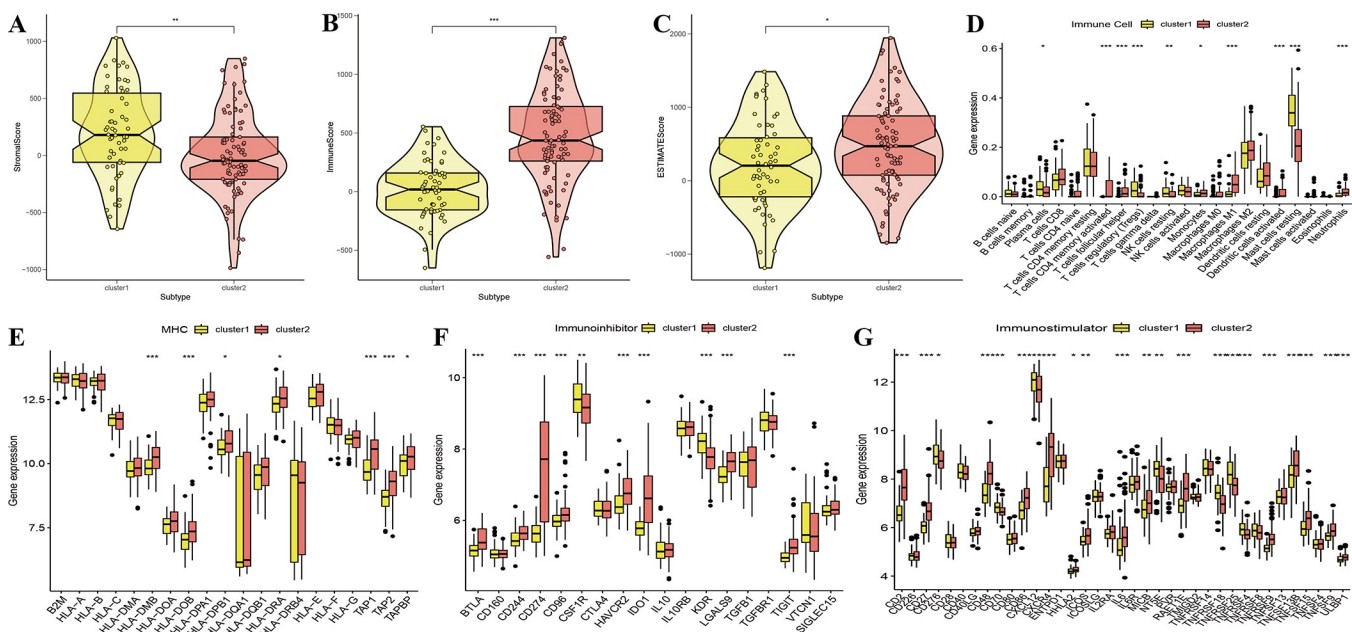

**Fig 8. Immune landscape between two subtypes.** (A-C) Comparison of stromal score, immune score and ESTIMATE score between different subtypes. (D) CIBERSORT analysis of immune cell infiltration in different subtypes. (E-G) Immune checkpoints analysis(MHC, immunoinhibitor, immunostimulator) in different subtypes. * P < 0.05, ** P < 0.001, and *** P < 0.0001.

treatments and prognoses. In our study, through mining the transcriptomic and proteomic data from psoriasis and healthy control skin tissues, PANoptosis-sig were identified using various bioinformatics techniques. We then constructed predictive and diagnostic models based on the PANoptosis-sig and evaluated their performance. In addition, associations between PANoptosis-sig and immune cells were analyzed using transcriptome and single-cell datasets. Finally, unsupervised consensus clustering analysis was conducted to ascertain psoriasis molecular subtypes. Immune landscape between different subtypes was comprehensively elaborated. These findings might provide valuable clues for the breakthrough of clinical diagnosis and treatment of psoriasis.

Five PANoptosis-sig (TYMP, S100A8, S100A9, NAMPT, LCN2) were identified through the integrated analysis of transcriptomic and proteomic data in our study. Previous research had found TYMP participated in drug metabolism-other enzyme pathways and played crucial roles in activating and proliferating immune cells in psoriasis vulgaris [39]. In a study evaluating the treatment of psoriasis with IL-17A and TNF-α inhibitors, it was observed that the differential expression of the protein TYMP before and after treatment was significant [40]. It had previously been found that the inflammation-associated proteins S100A8/9 were highly expressed in the lesional skin of patients with psoriasis [41]. S100A8/A9 might be involved in the development of high-risk coronary plaques in psoriasis [42].Through the regulation of complement factor C3, the S100A8/9 protein complex mediated psoriasis [43]. Keratinocytes were hyperproliferated and impaired in their terminal differentiation by intracellular NAMPT induced by Th1/Th17-cytokines [44]. Through parthanatos cell death, the NAMPT-derived nicotinamide adenine dinucleotide (NAD+) metabolism promoted skin inflammation [45]. In psoriasis, LCN2 modulated neutrophil activation and mediated skin inflammation [46]. By inhibiting the activation of the LCN2-dependent JAK/STAT pathway, miR-383 reduced keratinocyte proliferation and induced psoriasis apoptosis in a psoriasis animal study [47].

PANoptosis-sig played a crucial role in the pathogenesis of psoriasis at both the transcriptional and translational levels, making it a promising diagnostic markers and therapeutic targets.

In psoriasis, *IL-17* activated keratinocytes to produce chemokinesattracting neutrophilic granulocytes and *CCL20* attracting myeloid dendritic cells, which strengthened tissue inflammation [48–50]. When present with *IL-22* or *TNF-a*, *IL-17* induced the production of an antimicrobial protein that played a protective role when psoriasis lesions were combined with bacterial or fungal infections [51, 52]. We performed functional enrichment analysis on PANoptosis-sig, and the results showed enrichment in *IL−17* signaling pathway. Previous studies had shown interaction between *IL-17* and the adaptor protein *TRAF3IP2* promoted proliferation, inflammation, and inhibited apoptosis in epidermis cells [53]. The GO enrichment results showed the PANoptosis-sig were mainly enriched in response to biological process of leukocyte aggregation, leukocyte migration, chronic inflammatory response. The results suggested that PANoptosis might play an important role in the inflammatory response of psoriasis progression, and this possibility and mechanism needed to be further explored.

In this study, we conducted spearman correlation analysis between PANoptosis-sig and immune cells. Consistent with previous results [54], PANoptosis-sig were positively correlated with inflammatory infiltrating cells (macrophage, dendritic cell, *Th* cells, monocyte, neutrophil) and negatively correlated with fibroblasts and mast cells. Macrophage polarization was closely related to the pathogenesis of psoriasis, higher M1 polarization was associated with higher disease severity [55]. Macrophages in psoriatic arthritis synovium expressed markers of proinflammatory polarization and were mainly induced by macrophage colony-stimulating factors [56]. Research suggested that in early psoriasis, nucleic acids and various antimicrobial peptides released by damaged keratinocytes activated innate immune cells, including plasmacytoid DCs (pDCs) and macrophages, to produce tumor necrosis factor *TNF-α*, leading to the maturation of resident dermal DCs and the differentiation of monocytes into inflammatory DCs (iDCs) [57]. Mature resident DCs and rapidly increasing iDCs produced interleukins and other cytokines, strongly activating naive T cell differentiation. Keratinocytes also acted as immune cells by producing *TNF-α*, *IL-8*, vascular endothelial growth factor, antimicrobial peptides, some of which activate DCs. This vicious inflammatory cycle contributed to the persistence and worsening of plaques in the chronic phase of psoriasis [58, 59]. There was evidence that different subsets of *Th* cells participated in the inflammatory response of psoriasis [60]. The balance of *Th1*, *Th17*, and *Th2* cells was an important factor influencing the development of psoriasis [61]. Alefacept (anti-CD2) biologic therapy led to selective reduction of circulating effector memory T cells (Tem) and relative preservation of central memory T cells (Tcm) in psoriasis [62]. Zhang proposed resting mast cells were almost entirely absent in psoriatic skin. In contrast, the activated mast cells were enriched, suggesting that activation of mast cells might play a role in the pathogenesis of psoriasis [54]. Regulatory factors such as *IFN*, *IFN-γ* and *TNF* promoted the transformation of fibroblasts from fibrotic to inflammatory state in psoriasis lesions, and PANoptosis might play a role in this process [63]. In psoriasis, inflammatory fibroblasts contributed to the recruitment of *Th* cells and neutrophils [64]. Various immune cells played an indispensable role in the pathogenesis of psoriasis, and the interaction between PANoptosis and them remained to be further studied and explored.

Based on PANoptosis-sig, we divided psoriasis samples into two types by consensus clustering method. Then the difference of immune microenvironment between the two types was compared. The results displayed the immune and estimate scores of cluster 2 were expressively higher than those of cluster 1, and the expression of inflammatory immune cells (T cells, Macrophages, Dendritic cells activated, and Neutrophils) were also significantly higher than that of cluster 1. Interestingly, the expression of immune checkpoints were also significantly increased in cluster 2. These results suggested that cluster 2 was more sensitive to immunotherapy. The

molecular subtypes of psoriasis with PANoptosis-sig will help us achieve precision medicine in clinical diagnosis and provide patients with more personalized treatment strategies.

Bioinformatics approaches were being used in the current study to examine the transcriptional and translational landscape that PANoptosis sculpted in psoriasis. A broad range of advanced methodologies, including multiple machine learning algorithms, were employed in this thorough investigation. However it was important to acknowledge the presence of limitations, the discoveries lacked tests through molecular and animal experiments. Additionally, we will validate these findings next in the animal models and in the clinical cohort. These fundamentals may lead to novel diagnostic and therapeutic interventions by providing an in-depth understanding of psoriasis pathogenesis.

## Conclusion

In this study, the association between PANoptosis and psoriasis was investigated using novel bioinformatics approaches, including multiomics analysis, machine learning algorithms and consensus cluster analysis, with a specific focus on identifying PANoptosis-sig as potential biomarkers for psoriasis. Furthermore, we analyzed the relationship between PANoptosis-sig and immune cells in depth. These findings revealed potential biomarkers and therapeutic targets for the prevention, diagnosis, and treatment of psoriasis, enhancing our understanding of the molecular mechanisms underlying PANoptosis.

## Supporting information

**S1 Fig. Protein interaction analysis, Friends analysis and enrichment analysis of PDEGs.**
(TIF)

**S2 Fig. Reduction analysis in single-cell RNA sequencing.**
(TIF)

**S3 Fig. The expression of PANoptosis-sig in different cell populations.**
(TIF)

**S4 Fig. Validation of psoriasis subtypes and immune landscape between two subtypes in RNAseq datasets.**
(TIF)

**S1 Table. PANoptosis-related genes in GeneCards database.**
(XLSX)

**S2 Table. DEGs of psoriasis and healthy controls in GSE13355.**
(XLSX)

**S3 Table. DEPs of psoriasis and healthy controls.**
(XLSX)

**S4 Table. PANoptosis-related DEGs (PDEGs).**
(XLSX)

**S5 Table. PANoptosis-sig.**
(XLSX)

**S6 Table. DEGs between different subtypes in psoriasis.**
(XLSX)

**S1 Data.**
(ZIP)

## Acknowledgments

### Publisher's note

All claims expressed in this article are solely those of the authors and do not necessarily represent those of their affiliated organizations, or those of the publisher, the editors and the reviewers. Any product that may be evaluated in this article, or claim that may be made by its manufacturer, is not guaranteed or endorsed by the publisher.

## Author Contributions

**Conceptualization:** Qi Wang, Pei-feng He, Qi Yu.

**Data curation:** Li Wu, Xin-long Jiao, Ming Jing, Zhuo-yang Li, Ge-liang Liu, Kai Yan, Li-xuan Yan.

**Formal analysis:** Li Wu, Xin-long Jiao, Ming Jing, Chen-long Li, Gao-xiang Shi.

**Funding acquisition:** Yang Wang, Pei-feng He, Qi Yu.

**Methodology:** Li Wu, Xin-long Jiao, Ming Jing.

**Software:** Li Wu, Xin-long Jiao, Ming Jing.

**Supervision:** Qi Wang.

**Validation:** Li Wu, Xin-long Jiao, Ming Jing.

**Visualization:** Li Wu, Xin-long Jiao, Ming Jing, Chen-long Li, Gao-xiang Shi.

**Writing – original draft:** Li Wu, Xin-long Jiao, Ming Jing.

**Writing – review & editing:** Sheng-xiao Zhang, Yang Wang, Qi Wang, Pei-feng He, Qi Yu.

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
