## [Decision Letter · Decision Letter 0]

5 Jul 2024

PONE-D-24-19929Discovery of PANoptosis-related signatures correlates with immune cell infiltration in psoriasisPLOS ONE

Dear Dr. Yu,

Thank you for submitting your manuscript to PLOS ONE. After careful consideration, we feel that it has merit but does not fully meet PLOS ONE’s publication criteria as it currently stands. Therefore, we invite you to submit a revised version of the manuscript that addresses the points raised during the review process.

**There are major weaknesses in the methods as stated by the reviewers. The Figures' quality should be improved (pdf files are not sufficient). The Authors should clearly indicate what is new in their results as compared to those already published. **

If you choose to resubmit the manuscript, please submit your revised manuscript by Aug 19 2024 11:59PM. If you will need more time than this to complete your revisions, please reply to this message or contact the journal office at plosone@plos.org. Please include the following items when submitting your revised manuscript:A rebuttal letter that responds to each point raised by the academic editor and reviewer(s). You should upload this letter as a separate file labeled 'Response to Reviewers'.A marked-up copy of your manuscript that highlights changes made to the original version. You should upload this as a separate file labeled 'Revised Manuscript with Track Changes'.An unmarked version of your revised paper without tracked changes. You should upload this as a separate file labeled 'Manuscript'.

We look forward to receiving your revised manuscript.

Kind regards,

Michel Simon, Ph. D.

Academic Editor

PLOS ONE

Journal Requirements:

This project was supported by grants from Science and Technology Innovation of Shanxi Province (No. 202304051001017), the National Social Science Fund of China (No. 21BTQ050) and Shanxi Basic Research Project (No. 202203021212361).

4. Thank you for uploading your study's underlying data set. Unfortunately, the repository you have noted in your Data Availability statement does not qualify as an acceptable data repository according to PLOS's standards.

5. Please remove your figures from within your manuscript file, leaving only the individual TIFF/EPS image files, uploaded separately. These will be automatically included in the reviewers’ PDF.

Additional Editor Comments:

The aim of this study was to re-analyze psoriasis RNAseq data with newly developped bioinformatics tools in order to test a possible involvement of PANoptosis-related mechanisms in the disease pathogenesis. Such a study could be of interest for understanding the disease and proposition of new therapeutic targets.

However, the quality of figures are not sufficient to evaluate the accuracy of the reported data. The methods were not sufficently described.

I did not not understand the differences between the submitted data (Fig 1) and the already published ones starting from the same GSE13355 dataset (Clin Cosmet Invest Dermatol, 2024 17:1281-95). Differential expression was considered as significant when |log2FC|>1.5, as stated in the methods section. This is apparently not the case on Fig 1A (volcano plot). Please clarify.

33 genes compose the panoptosis signature (GeneCards data base). Did the Authors considered these genes? If not (IFI6, CD274, S100A8/9, 62MB were apparently not), please explain.

The abstract is not informative about the results, it sounds like a brief description of the methods used.

What do your analysis add to the understanding of the disease?

Reviewers' comments:

Reviewer's Responses to Questions

**Comments to the Author**

1. Is the manuscript technically sound, and do the data support the conclusions?

Reviewer #1: Partly

Reviewer #2: Partly

2. Has the statistical analysis been performed appropriately and rigorously? 

Reviewer #1: No

Reviewer #2: No

3. Have the authors made all data underlying the findings in their manuscript fully available?

Reviewer #1: No

Reviewer #2: Yes

4. Is the manuscript presented in an intelligible fashion and written in standard English?

Reviewer #1: Yes

Reviewer #2: Yes

5. Review Comments to the Author

**Reviewer #1:** 1. This study used quite small number of samples. Why only microarray dat were used? Pleanty of RNAseq data (GSE66511) are available and using microarrays seems strange.

2. How the data were combined? The datasets taht outhors use are from different platforms and cannot be joined.

3. Statistical signficance table is missing? WHat genes were differentialy expressed.

4 The authors should use this tool https://geoexplorer.rosalind.kcl.ac.uk . it helps to combine several datasets from different plaforms and helps to incraese the power.

5. The abstract states that "These findings revealed potential biomarkers and therapeutic targets for the prevention, diagnosis, and treatment of psoriasis, enhancing our understanding of the molecular mechanisms underlying PANoptosis." What are these genes ? The authors do not present any genes that are differnetially expressed.

6. It is not understandable, how this study is different from any differential gene expression analysis? What are the new finidngs and new results

7. The quaity of figures is low and they are illegible.

8. Overall, the sample size is small for this type of study. Sample size is so small that it excludes efficient splitting it into training and experimental sets.

The major reviison is required, addiiton of RNAseq data or even using only RNAseq data. More description in the methidos and gene expreeison tables have to be presented.

**Reviewer #2:** The manuscript entitled “Discovery of PANoptosis-related signatures correlates with immune cell infiltration in psoriasis” offered new insights into potential PANoptosis-related mechanisms in psoriasis, It includes acquisition of differentially expressed genes, recognition of PANoptosis-sig, construction of diagnostic prediction models and performance evaluation, transcriptome analysis, single-cell datasets analysis, cluster analysis. Enhanced understanding of the molecular mechanisms of PANoptosis. The manuscript needs revisions, as explained below:

Comments:

-1.How does single-cell sequencing address the issue of technical variability, including the exclusion of false positive or false negative results?

-2.What are the important components of PANoptosome in the Introduction? Caspase-1 and Caspase-8 are not mentioned?

-3.It is suggested to add PCR and immunohistochemical experiments for verification.

-4.How to remove batch differences between different data sets in the article?

-5.This paper analyzes the results from the genomics and is it possible to analyze the results from the proteomics to enhance the feasibility of the results.

-6.Please summarize the main findings of the study.

-7.At the end of the Introduction is not clear which are the innovative aspects of the research.

6. PLOS authors have the option to publish the peer review history of their article (what does this mean?). If published, this will include your full peer review and any attached files.

Reviewer #1: No

Reviewer #2: No

---

## [Author Response · Author response to Decision Letter 0]

18 Aug 2024

Dear reviewers: 

Thank you very much for your careful review of our manuscript entitled “Discovery of PANoptosis-related signatures correlates with immune cell infiltration in psoriasis”. We have carefully considered the reviewers’ professional comments and made a point to point response as follows: 

To Reviewer 1： 

1. This study used quite small number of samples. Why only microarray data were used? Pleanty of RNAseq data (GSE66511) are available and using microarrays seems strange.

Response：We appreciate your insightful comments regarding our manuscript. We acknowledge the limitations associated with using only microarray data in our study. In response to this concern, we have included RNAseq data from the GSE66511, GSE54456 and GSE121212 datasets, as suggested.

It should be noted that the inclusion criteria for our study necessitated datasets containing both psoriatic and healthy control skin samples, with each group comprising no fewer than 20 samples of skin tissue. Consequently, our final dataset encompassed 276 psoriatic lesion skin samples and 247 healthy control skin samples. This comprehensive datasets now allowed for a more robust analysis, leveraging the strengths of both microarray and RNAseq technologies. We believe that this addition significantly enhances the rigor and scope of our study, addressing your valid concern regarding the choice of data type. We are grateful for the opportunity to improve our work based on such constructive feedback.

2. How the data were combined? The datasets that authors use are from different platforms and cannot be joined.

Response：We appreciate your query regarding the integration of datasets from different platforms in our study. To address this challenge, we employed the SVA (Surrogate Variable Analysis) package within the R programming environment for data integration. SVA is a well-established method for integrating heterogeneous datasets, such as those derived from microarray and RNAseq platforms[1]. This approach involves modeling and adjusting for batch effects and other sources of variability that arise from different experimental platforms. Specifically, SVA utilizes surrogate variables to capture and adjust for latent sources of variation that are not directly related to the biological variables of interest, thereby harmonizing the datasets prior to downstream analyses.

In our study, after preprocessing the microarray and RNAseq data separately according to their respective platform-specific protocols, we applied SVA to adjust for technical variations and ensure comparability across datasets. This methodological step allowed us to effectively combine the datasets while controlling for potential biases introduced by platform-specific differences. Kwanha Yu also used this approach to remove batch effect in their study[2]. Shi Zhang used SVA to merge microarray datasets and decrease heterogeneities in their studies[3]. Andrew E Teschendorff described the reliability that SVA could be used to eliminate confounders in large-scale microarray analysis[4]. So, We believe that employing SVA for data integration strengthens the validity of our findings by minimizing the impact of technical variability and maximizing the biological insights gained from the integrated analysis.

References:

[1]. Leek JT, Johnson WE, Parker HS, Jaffe AE, Storey JD. The sva package for removing batch effects and other unwanted variation in high-throughput experiments. Bioinformatics (Oxford, England). 2012;28(6):882-3.

[2]. Yu K, Lin CJ, Hatcher A, Lozzi B, Kong K, Huang-Hobbs E, et al. PIK3CA variants selectively initiate brain hyperactivity during gliomagenesis. Nature. 2020;578(7793):166-71.

[3]. Zhang S, Wu Z, Xie J, Yang Y, Wang L, Qiu H. DNA methylation exploration for ARDS: a multi-omics and multi-microarray interrelated analysis. Journal of translational medicine. 2019;17(1):345.

[4]. Teschendorff AE, Zhuang J, Widschwendter M. Independent surrogate variable analysis to deconvolve confounding factors in large-scale microarray profiling studies. Bioinformatics (Oxford, England). 2011;27(11):1496-505.

3. Statistical significance table is missing? What genes were differentialy expressed.

Response：Thank you for your professional suggestions. We apologized for the previous omission of not providing the results of differential gene expression analysis. The results have now been added to the supplementary table(S2 Table) in the latest version of the manuscript.

To conduct differential gene expression analysis, we employed the limma (Linear Models for Microarray Data) package, a widely used tool for identifying differential expressed genes (DEGs). Limma fit linear models to expression data from microarray experiments and RNAseq, allowing for robust statistical inference even with limited sample sizes. In our study, we applied limma to analyze gene expression differences between psoriatic lesion and healthy control samples in the training set. We adhered to standard procedures recommended by the limma package, including normalization, correction for multiple testing, and the determination of fold changes and associated statistical significance levels (adjusted p-values). The results of our differential expression analysis, a total of 256 DEGs were identified from psoriasis and healthy control samples, with 191 genes upregulated and 65 genes downregulated (Fig 1A, S2 Table). The S2 table provided a comprehensive view of the molecular changes associated with psoriasis pathogenesis, highlighting genes which were significantly upregulated or downregulated in psoriatic lesions compared to healthy skin.

4. The authors should use this tool https://geoexplorer.rosalind.kcl.ac.uk . it helps to combine several datasets from different platforms and helps to increase the power.

Response：Thank you for your insightful suggestion regarding the use of the GEOexplorer tool for combining datasets. We appreciate your recommendation and understand its potential to enhance data integration and analysis. In our study, we opted to utilize the SVA package in R for data merging and correction of unrelated variation. This choice was based on our familiarity with the package's capabilities and specific advantages in our analysis context. However, we acknowledge the benefits of exploring alternative tools like GEOexplorer, and we will consider incorporating insights from this platform into our future work. Your feedback is invaluable in guiding us toward more robust methodologies.

5. The abstract states that "These findings revealed potential biomarkers and therapeutic targets for the prevention, diagnosis, and treatment of psoriasis, enhancing our understanding of the molecular mechanisms underlying PANoptosis." What are these genes ? The authors do not present any genes that are differnetially expressed.

Response：Thank you for your professional suggestions. In our study, we had revised our analysis methods based on the feedback from the editor and reviewers, resulting in new PANoptosis signatures (PANoptosis-sig). This information had been included in the abstract of the revised version. We identified five key PANoptosis-sig that might serve as potential biomarkers and therapeutic targets for psoriasis. These genes were TYMP, S100A8, S100A9, NAMPT and LCN2. We validated the differential expression of PANoptosis-sig between psoriasis and healthy controls at transcriptional and protein levels. The enrichment analysis highlighted the roles of PANoptosis-sig in inflammation-related processes such as leukocyte aggregation, migration, and responses to the IL-17 signaling pathway. Overall, PANoptosis-sig represented potential targets for future therapeutic interventions and diagnostic tools in managing psoriasis, thereby enhancing our understanding of the molecular mechanisms contributing to the disease.

6. It is not understandable, how this study is different from any differential gene expression analysis? What are the new findngs and new results.

Response：Thank you for your professional suggestions. We would like to emphasize that our research integrated multi-omics data from transcriptomics, proteomics, and single-cell transcriptomics. We employed a variety of bioinformatics approaches, including differential gene expression analysis, Weighted Gene Co-Expression Network Analysis (WGCNA), and machine learning techniques to identify biomarkers associated with PANoptosis in psoriasis. Furthermore, our study presented a novel molecular classification of psoriasis patient cohorts based on PANoptosis signature (PANoptosis-sig). This classification had been validated in our testing cohort, demonstrating reliability and stability. Significant differences in immune landscape among the distinct molecular subtypes were represented, aiding in personalized treatment strategies for psoriasis patients.

We believe these contributions represent a significant advancement in understanding the molecular underpinnings of psoriasis and can have meaningful implications for clinical practice. 

7. The quality of figures is low and they are illegible.

Response：Thank you for your valuable feedback regarding the quality of the figures in our manuscript. We acknowledge that the low resolution and illegibility detracted from the clarity of our findings. In response to your concerns, we will upload high-resolution versions of all figures to ensure they are clear and legible. We understand that high-quality visuals are essential for effectively communicating our data and supporting our conclusions. Thank you for your understanding and for giving us the opportunity to improve our submission.

8. Overall, the sample size is small for this type of study. Sample size is so small that it excludes efficient splitting it into training and experimental sets.

Response：Thank you for your professional suggestions. We acknowledge the concern raised about the initial sample size being insufficient for splitting into training and experimental sets. To address this limitation, we have expanded our dataset to include additional RNAseq data. The revised study now incorporates a total of 5 microarray datasets and 3 RNAseq datasets, comprising 276 psoriatic skin samples and 247 healthy control samples. Specifically, we had designated the GSE13355 dataset as the training set, while the remaining datasets had been utilized as validation sets. This approach enhanced the robustness of our model by ensuring a more reliable and generalized evaluation of our findings. For a detailed breakdown of the datasets and their allocation, please refer to Table 1 in the methods section of the revised manuscript. We believe this increased sample size will significantly improve the reliability and validity of our results, addressing the initial concerns regarding the dataset's adequacy for robust model training and validation.

9. The major revision is required, addition of RNAseq data or even using only RNAseq data. More description in the methods and gene expression tables have to be presented.

Response：We appreciate your valuable suggestions. We have incorporated RNAseq data analysis into our revised manuscript to corroborate our research findings. The inclusion of RNAseq data allowed us to validate our initial findings with a more robust approach, thereby strengthening the scientific rigor and reliability of our conclusions. Furthermore, we have provided expanded methodological details in the methods section and included gene expression tables in supplementary materials(https://www.jianguoyun.com/p/DZjVNUoQkbumDBiVlK4FIAA) to enhance the transparency and reproducibility of our study.

To Reviewer 2：

1. How does single-cell sequencing address the issue of technical variability, including the exclusion of false positive or false negative results?

Response：Thank you for your insightful question regarding the management of technical variability and the handling of false positive and false negative results in single-cell sequencing. The single-cell dataset(GSE151177) used in our study uses 10×Genomics technology. The 10×Genomics single-cell sequencing technology incorporates several methodologies to mitigate these issues. 

Firstly, 10×Genomics single-cell sequencing utilizes a microfluidic-based approach to encapsulate individual cells into unique droplets along with barcoded beads. Each bead is pre-coated with unique oligonucleotides that are designed to capture cDNA from mRNA molecules in the cell. This approach minimizes cross-contamination between cells and ensures that each transcript is accurately associated with its cell of origin. Additionally, the integration of multiplexed bead-based technology enhances the reproducibility of results by reducing technical variability. 

Secondly, to address false positives, the 10×Genomics platform employs several layers of quality control. First, the system includes stringent criteria for filtering out low-quality cells based on metrics such as the number of detected genes and the total RNA content. Cells with unusually high levels of detected genes or excessive mitochondrial gene expression are flagged as potential technical artifacts. Furthermore, the use of barcoding and unique molecular identifiers (UMIs) helps in distinguishing true biological signals from technical noise by enabling the accurate counting of unique transcripts, thereby reducing the impact of amplification artifacts. In our research, data quality control was performed separately for each sample before data integration. In order to eliminate partial cells and doublets, genes expressed in fewer than three cells, cells with more than 5,000 genes, and mitochondrial genes expressed in more than 25% by mitochondrial cells were filtered out.

Thirdly, to minimize false negatives, the 10×Genomics protocol employs a comprehensive approach to capture a broad range of transcripts, including those present in low abundance. The use of UMIs also aids in accurately quantifying lowly expressed genes by distinguishing between genuine biological signals and technical dropouts. Additionally, the high sensitivity of the microfluidic encapsulation method ensures that even rare cell populations are captured and analyzed, thus reducing the likelihood of missing important biological information.

Overall, the 10×Genomics single-cell sequencing platform combines advanced technical features with robust quality control measures to address the challenges associated with technical variability and to minimize the occurrence of false positive and false negative results. These strategies collectively enhance the reliability and accuracy of single-cell transcriptomic analyses.

2. What are the important components of PANoptosome in the Introduction? Caspase-1 and Caspase-8 are not mentioned?

Response：Thank you for your professional suggestions. We have modified the relevant content of PANoptosome in the artical, please refer to the introduction part of the manuscript for details. PANoptosome complexes function as molecular scaffolds for concurrently engaging multiple key molecules from the pyroptotic (NLRP3, ASC, Caspase-1), apoptotic(Caspase-8, Caspase-3, FADD), and necroptotic (RIPK1 and RIPK3) cell death pathways[5-9]. Genetic and biochemical evidence indicates that proteins within the PANoptosome can be broadly divided into three groups: (1) ZBP1 and NLRP3, which are likely PAMP and DAMP detectors, (2) ASC and FADD, serving as adaptors, and (3) CASP1, CASP8, RIPK1 and RIPK3, as catalytic effectors[10]. PANoptosis is a process with intermolecular interaction among pyroptosis, apoptosis, and necroptosis[11]. PANoptosome complexes exert scaffolds and enzyme-based catalytic modifications to induce PANoptosis. 

References:

[5]. Zheng M, Kanneganti TD. The regulation of the ZBP1-NLRP3 inflammasome and its implications in pyroptosis, apoptosis, and necroptosis (PANoptosis). Immunological reviews. 2020;297(1):26-38.

[6]. Lee S, Karki R, Wang Y, Nguyen LN, Kalathur RC, Kanneganti TD. AIM2 forms a complex with pyrin and ZBP1 to drive PANoptosis and host defence. Nature. 2021;597(7876):415-9.

[7]. Briard B, Malireddi RKS, Kanneganti TD. Role of inflammasomes/pyroptosis and PANoptosis during fungal infection. PLoS pathogens. 2021;17(3):e1009358.

[8]. Malireddi RKS, Kesavardhana S, Karki R, Kancharana B, Burton AR, Kanneganti TD. 

---

## [Decision Letter · Decision Letter 1]

30 Aug 2024

Discovery of PANoptosis-related signatures correlates with immune cell infiltration in psoriasis

PONE-D-24-19929R1

Dear Dr. Qi Yu,

We’re pleased to inform you that your manuscript has been judged scientifically suitable for publication and will be formally accepted for publication once it meets all outstanding technical requirements.

Kind regards,

Masanori A. Murayama

Academic Editor

PLOS ONE

Additional Editor Comments (optional):

Thank you for submitting your revised manuscript. We are pleased to inform about decision. You have addressed all comments from reviewers, and they endorsed the publish..

Reviewers' comments:

Reviewer's Responses to Questions

**Comments to the Author**

1. If the authors have adequately addressed your comments raised in a previous round of review and you feel that this manuscript is now acceptable for publication, you may indicate that here to bypass the “Comments to the Author” section, enter your conflict of interest statement in the “Confidential to Editor” section, and submit your "Accept" recommendation.

Reviewer #1: All comments have been addressed

Reviewer #2: All comments have been addressed

2. Is the manuscript technically sound, and do the data support the conclusions?

Reviewer #1: Yes

Reviewer #2: Yes

3. Has the statistical analysis been performed appropriately and rigorously? 

Reviewer #1: Yes

Reviewer #2: Yes

4. Have the authors made all data underlying the findings in their manuscript fully available?

Reviewer #1: Yes

Reviewer #2: Yes

5. Is the manuscript presented in an intelligible fashion and written in standard English?

Reviewer #1: Yes

Reviewer #2: Yes

6. Review Comments to the Author

Reviewer #1: I do not have any comments I do not have any comments I do not have any comments I do not have any comments

Reviewer #2: I do not have other concerns, the authors addressed all my concerns, thank you for your revision, good luck.

7. PLOS authors have the option to publish the peer review history of their article (what does this mean?). If published, this will include your full peer review and any attached files.

Reviewer #1: No

Reviewer #2: No

---

## [Editor Report · Acceptance letter]

2 Sep 2024

PONE-D-24-19929R1 

PLOS ONE

Dear Dr. Yu, 

I'm pleased to inform you that your manuscript has been deemed suitable for publication in PLOS ONE. Congratulations! Your manuscript is now being handed over to our production team.

Kind regards, 

on behalf of

Dr. Masanori A. Murayama 

Academic Editor

PLOS ONE